# The Role of Magnetic Resonance Enterography in Crohn’s Disease: A Review of Recent Literature

**DOI:** 10.3390/diagnostics12051236

**Published:** 2022-05-15

**Authors:** Marysol Biondi, Eleonora Bicci, Ginevra Danti, Federica Flammia, Giuditta Chiti, Pierpaolo Palumbo, Federico Bruno, Alessandra Borgheresi, Roberta Grassi, Francesca Grassi, Roberta Fusco, Vincenza Granata, Andrea Giovagnoni, Antonio Barile, Vittorio Miele

**Affiliations:** 1Department of Radiology, Careggi University Hospital, Largo Brambilla 3, 50134 Florence, Italy; marysolbiondi@gmail.com (M.B.); eleonora.bicci92@gmail.com (E.B.); federicaflammia91@gmail.com (F.F.); giudittachiti@gmail.com (G.C.); vmiele@sirm.org (V.M.); 2Abruzzo Health Unit 1, Department of Diagnostic Imaging, Area of Cardiovascular and Interventional Imaging, 67100 L’Aquila, Italy; palumbopierpaolo89@gmail.com; 3Department of Biotechnological and Applied Clinical Sciences, University of L’Aquila, 67100 L’Aquila, Italy; federico.bruno.1988@gmail.com (F.B.); abarile63@gmail.com (A.B.); 4Department of Clinical Special and Dental Sciences, School of Radiology, University Politecnica delle Marche, 60126 Ancona, Italy; a.borgheresi@staff.univpm.it; 5Division of Radiology, Università degli Studi della Campania Luigi Vanvitelli, 81100 Naples, Italy; roberta.grassi@policliniconapoli.it; 6Department of Precision Medicine, University of Campania “L. Vanvitelli”, 81100 Naples, Italy; francescagrassi1996@gmail.com; 7Medical Oncology Division, Igea SpA, 80013 Naples, Italy; r.fusco@igeamedical.com; 8Division of Radiology, Istituto Nazionale Tumori IRCCS Fondazione Pascale-IRCCS di Napoli, 80131 Naples, Italy; v.granata@istitutotumori.na.it; 9Departement of Radiological Sciences, Ospedali Riuniti Ancona, University Hospital, Università Politecnica delle Marche, Via Tronto 10, 60126 Ancona, Italy; a.giovagnoni@univpm.it

**Keywords:** inflammatory bowel disease, Crohn’s disease, imaging, MR enterography, disease activity, complications

## Abstract

Inflammatory bowel disease (IBD) is the term used to identify a form of chronic inflammation of the gastrointestinal tract that primarily contemplates two major entities: ulcerative colitis (UC) and Crohn’s disease (CD). The classic signs are abdominal pain and diarrhoea that correlate with the localization of gastro-enteric disease, although in this pathology extraintestinal symptoms may coexist. The diagnosis of CD relies on a synergistic combination of clinical, laboratory (stool and biochemical), cross-sectional imaging evaluation, as well as endoscopic and histologic assessments. The purpose of this paper is to prove the role of imaging in the diagnosis and follow-up of patients with CD with particular focus on recent innovations of magnetic resonance enterography (MRE) as a pivotal diagnostic tool, analysing the MRE study protocol and imaging features during the various phases of disease activity and its complications.

## 1. Introduction

Inflammatory bowel disease (IBD) represents a group of chronic inflammation of the gastrointestinal tract, which mainly comprises two main forms: ulcerative colitis (UC) and Crohn’s disease (CD) [1,2]. CD is a disorder with multifactorial aetiology in which genetics and environment are deeply involved in determining the manifestation of the disease. In fact, risk factors include genetic determinants (so far, the most significant genetic associations have been found in NOD2, IL23R, and ATG16L1 genes) and acquired determinants, such as a diet low in carbohydrates, smoking, use of non-steroidal anti-inflammatory drug (NSAIDs) and an altered intestinal microbiome [3,4,5]. The manifestations of CD and UC are similar, although in UC the patient more commonly presents with diarrhoea and bleeding, whereas the patient with CD typically presents with watery diarrhoea and more nuanced symptoms [6]. CD can affect with varying degrees of severity any part of the gastrointestinal (GI) tract, from the oral cavity to the anus, so clinical manifestations can include a wide spectrum of possibilities depending on location and severity of disease [7,8]. Abdominal pain is usually discrete in magnitude and frequently precedes diagnosis of several years; it is associated with bowel movements and is usually localised in the right lower quadrant by virtue of the preferential localization of the disease at the level of the terminal ileum. Diarrhoea is generally watery but may be bloody, especially if the colorectal tract is involved [9]. Weight loss is due to multiple factors, such as chronic diarrhoea, malabsorption, and not least anorexia resulting from fear of eating. In CD, UC can also be present, with a prevalence between 21% and 47%, and several typical extraintestinal manifestations. The most common are the skin ones (pyoderma gangrenosum, erythema nodosum and enterocutaneous fistulae), followed by hepato-biliary, musculoskeletal, genitourinary, respiratory, ocular, and cardiovascular manifestations [10,11]. As stated by recent European Society of Gastrointestinal and Abdominal Radiology (ESGAR) guidelines, the diagnosis of CD does not contemplate a single reference standard, but rather relies on a synergistic combination of clinical, laboratory (stool and biochemical), and cross-sectional imaging evaluation, as well as endoscopic and histologic assessment [12,13,14,15,16,17,18,19,20]. Ileocolonoscopy with biopsies from inflamed and healthy intestinal segments is essential to establish the diagnosis in cases of suspected IBD and the most significant endoscopic features are the presence of discontinuous lesions, stenosis, and fistulae with perianal involvement [21,22,23,24]. Small bowel capsule endoscopy (SBCE) or cross-sectional imaging indeed should be considered in patients with clinical suspicion of CD and normal endoscopy [25,26]. In addition, all patients with newly diagnosed CD should undergo evaluation of the small bowel by bowel ultrasound (US), magnetic resonance enterography (MRE), and/or capsule endoscopy [27,28]. With regard to treatment assessment, today, there is no reference standard: clinical symptoms as scored by the CD Activity Index [CDAI] are not a reliable measure of underlying inflammation, while a growing number of studies suggest that mucosal healing (MH), which can be visualised directly by endoscopy, can change the natural course of CD by decreasing relapse rates, hospitalisation rates, and the need for surgery [29,30,31]. Again, another parameter that allows us to evaluate the response to therapy is the transmural healing (MH), because CD is a transmural pathology and the full-thickness healing of the tract affected by pathology with restoration of a normal wall thickness can be an important point for the evaluation of the response. In this context, any type of diagnostic modality (including endoscopy, MRE, laboratory parameters, or US) can only be used as a surrogate marker for evaluating transmural healing. Numerous studies have corroborated the role of cross-sectional imaging techniques (US, computed tomography enterography (CTE), or MRE) for therapeutic monitoring in CD affecting the small and large intestines [32,33,34,35,36,37,38,39]. While the ileocecal region is usually visualised effectively endoscopically, the proximal ileum and jejunum are more difficult to evaluate. In order to meet the need for an exhaustive evaluation of the entire gastro-enteric tract with particular emphasis on the small bowel, CTE and MRE are used to determine the extent and activity of the disease based on wall thickness and intravenous contrast enhancement [28]. Both techniques have high sensitivity and specificity. However, due to the absence of radiation, MRE should be preferable to CTE, particularly in young patients [40,41,42,43,44,45,46,47,48]. In relation to what has been stated so far, it follows that imaging has an unquestionable role in the diagnosis and post-treatment assessment of the disease, with particular effectiveness in the evaluation of small bowel lesions. Among the various techniques that can be used is MRE, playing a predominant role due to its panoramic nature, absence of ionising radiation, excellent contrast resolution, and the multiparametric nature of its inherent features [39].

## 2. Role of Imaging

Radiological imaging, and in particular US, CTE, and MRE, is becoming increasingly important both in the diagnosis and follow-up of CD patients. At the time of diagnosis, radiology has a fundamental role in assessing the extent of disease and the presence of complications, such as fistulas, abscesses, or strictures, as well as the state of disease activity. In the follow-up, especially, assessment of disease activity is mandatory to evaluate response to therapy and to detect complications [49,50,51,52].

## 3. Ultrasound Assessment

US is a non-invasive method that requires no preparation other than fasting in the hours prior to the examination, to reduce the presence of intestinal meteorism that could potentially interfere with this one. It does not use ionising radiation and is therefore especially useful in young patients. Normally, the intestinal wall has five concentric layers with alternating hypo- and hyper-echogenic layers, in particular starting from the intestinal lumen: mucosa, deep mucosa, muscularis mucosae, sub-mucosa, muscularis propria and serosa. Under physiological conditions, the intestinal wall is very thin with poor signal on completion with colour, i.e., Doppler [53,54,55]. One of the features that is often seen at US in IBD is wall thickening. A wall thickening of >3 mm is identified as a cut-off for pathology. In CD, there is usually a thickness of 5–15 mm depending on the state of disease activity. Thickening is associated with the loss of normal wall stratification. In addition to this, evaluation by colour Doppler or contrast-enhanced ultrasound (CEUS) may be useful in assessing wall hyperemia secondary to the inflammatory state. The presence of increased wall vascularity is a sign of disease activity. CEUS can assess wall perfusion and therefore also in this case, the inflammatory status [56,57,58,59]. US may also reveal alterations in peristalsis, with a decrease in comparison to normal intestinal loops especially in cases of fibrosis or increased peristalsis of loops located before stenosis [60,61]. Other findings are the presence of thickening of peri-visceral adipose tissue with lymphadenopathy [62]. US also provides support in the case of transmural complications. The most common of these are sinus tracts, fistulae, inflammatory masses, or abscesses. As the disease worsens, the inflammation deepens in the wall, creating interruptions in its continuity [63]. This communication with the intestinal lumen may continue into the peri-visceral fat which can be visualised as small hypoechogenic tracts within the hyperechogenic fat. The fistula will form when the sinus tract connects the involved loop with another organ or with the cutaneous surface. Abscesses and inflammatory masses are often located near the site of disease and are often associated with fistulas [64]. They present on US as hypo-hyper echogenic masses with increased vascularity on colour Doppler. The key difference on US between these two entities is the liquid content without enhancement on colour Doppler of the abscess compared with the inflammatory mass [65].

## 4. Computed Tomography Assessment

CT is an alternative imaging to US and MRI which is often used for the evaluation of CD due to its simple execution and rapidity [66,67,68]. In particular, there are cases where it becomes the imaging of choice, especially in emergency, post-operative settings, in patients with allergy to gadolinium-based contrast agents, or in patients who do not tolerate long-term examinations such as MRE. This is performed with a combination of one litre of oral fluid (neutral or low density) with subsequent administration of intravenous iodinated contrast medium, in order to optimise the distension of the lumen of the loops and the wall enhancement [69]. However, given the often young age of the patients and the use of ionising radiation, protocols using low-dose CT and iterative reconstruction algorithms were evaluated [69,70,71]. Some studies showed that low-dose CTE using model-based-iterative reconstruction (MBIR) is specific and sensitive in detecting CD, but using lower doses of ionising radiation than the standard protocol [72]. With the same aim of reducing the radiation dose delivered to the patient, studies on the potential use of dual-energy CT (DECT) have also been conducted. The inflammatory state is closely related to an elevated blood flow to the involved organ, and therefore to an increased vascularisation and a higher concentration of iodinated contrast medium once it has been administered. Thus, the iodine map on DECT could indicate the activity of CD. These maps, combined with an optimised kiloelectron volt (keV) for virtual monoenergetic imaging (VMI), may allow a correct diagnosis of CD with dose reduction [73,74,75]. In addition, the ever-increasing interest in radiomics and artificial intelligence is also leading to many studies on CD, conducted on CTE and MRE imaging, both in the assessment of the disease state, and in particular the presence of fibrosis, and in the differential diagnosis between CD and other inflammatory bowel diseases such as UC [76,77,78,79,80,81,82,83,84,85,86].

## 5. Magnetic Resonance Enterography

CTE and MRE are nowadays recognized as widely accepted methods for the proper and accurate evaluation of the small intestine in patients with CD [87,88,89,90,91,92,93,94,95,96,97,98,99,100,101]. In consideration of its innumerable and unquestionable advantages, MRE stands out among the methods that can be used in the study of CD and we can affirm that today it represents the gold standard method. Among the unquestionable advantages of MRE, first of all, there is the absence of ionising radiation that allows its use both in pregnant patients, in paediatric subjects, and in the follow-up of a patient with diagnosed disease [42].

## 6. Protocols

Image quality is essential to achieve high diagnostic accuracy. To this end, it is paramount to obtain adequate bowel distension and to select the most suitable sequences to obtain a quality study.

The ESGAR/ESPR consensus statement provides useful recommendations, describing a standardised approach to patient preparation and acquisition protocols for MRE, to guide cross-sectional radiological practice for small bowel and colon imaging [102].

## 7. General Patient Preparation

There is little evidence on optimal patient preparation prior to MRE and recommendations for fasting periods for solids and fluids are mainly based on expert opinion. Expert opinion concerning periods of nil by mouth for solids recommended that patients should not eat any solid for 4–6 h [103]. The intake of sparkling water is not recommended because of the risk of producing intraluminal gas artefacts on MRE. There is no consensus regarding bowel preparation, although there is evidence that a clean colon facilitates the transit of intraluminal contrast and avoids reflux of faecal material into the distal ileum. So, the routine use of bowel laxatives and rectal enema is not recommended, but the administration of a water enema in comparison to evaluating the un-prepared colon, when not contraindicated, improves detecting colonic inflammation with MRE [103,104]. Even if no consensus was reached, recent studies have shown that the use of spasmolytic agents significantly improves loop distension and therefore the use of spasmolytic prior to MRE is recommended, if not contraindicated [105]. Hyoscine butylbromide is the recommended first-line agent at the dose of 20 mg. Either a single dose or a fractionated dose is appropriate. Second line agent is i.v. glucagon and the recommended dose is 1 mg. Spasmolytic agent’s administration is most effective when given intravenously and should precede to motion-sensitive sequences (typically fast spin echo T2-weighted sequence and post contrast T1-weighted images) [106]. There is good evidence that MRI accuracy is improved by oral contrast administration [107,108]. In some studies, better distension of the small bowel has been shown to be achieved with enteroclysis than with routine oral administration of contrast [109,110], but no significant impact on clinical decision making has been demonstrated. All in all, enteroclysis is not considered necessary for a valid MRE evaluation of the small bowel in inflammatory bowel disease by either the United States or the European consensus guide. Enteroclysis involves first inserting a naso-jejunal tube via fluoroscopic guidance with the distal end of the tube into the proximal jejunum to prevent the patient from vomiting. The contrast agent is then introduced through the tube, and introduction can be manual or with an automated pump [111]. The entire procedure is performed under fluoroscopic guidance, so it involves radiation exposure for the patient and is not desirable in paediatric patients. In addition, the procedure overall is a major discomfort for the patient even though it allows for effective distension of the ileum and jejunum. A valuable alternative to the enteroclysis is MRE in which contrast medium is introduced per OS. The procedure is easier to perform and better tolerated by the patient, although it results in reduced distension of the jejunal loops compared with enteroclysis [112,113]. Although there is evidence that reasonable quality examinations can be obtained with as little as 450 mL of oral contrast medium, the optimal volume of oral contrast is 1000–1500 mL [114,115]. In unoperated patients, it should be ingested over 45–60 min before the examination, whereas in patients with previous major small bowel resection, scanning earlier, e.g., at 30 min, may be advisable [116]. It is recommended that when scanning patients with a stoma, the stoma should be plugged before ingesting oral contrast. There is no single preferred agent for MRE, although recommended agents include biphasic agents such as mannitol, PEG, sorbitol, and lactulose among others [117,118,119]. Among biphasic agents, water can be used, but it has the limitation of being absorbed rapidly and thus does not provide long-lasting loop relaxation. More frequently, PEG-based, polyethylene glycol compounds can be used, which provide more effective and long-lasting distension.

## 8. Technical Considerations and Sequence Selection

The scan includes small bowel and colon from diaphragm to perineum and the total acquisition time should be equal or <30 min. Imaging in the prone position is preferred when possible because it appears to decrease artefacts from respiratory motion, improve luminal distension, reduce the time to acquire the coronal sequence by compressing the abdomen, and be more tolerable for patients suffering from claustrophobia. However, there is no evidence that it improves diagnostic adequacy [120]. In addition, some patients find this position difficult, particularly those with a stoma, so supine positioning is considered to be acceptable.

The basic set of acquisition sequences recommended for CD by both European and US guidelines includes (Table 1):Axial and coronal T2 FSE without FS 2DAxial and coronal SSFPGE without FSAxial or coronal T2 FSE with FS 2DAxial and coronal pre- and post- contrast 3D T1-weighted gradient-echo sequence with FS

Optional sequences:
Axial Diffusion Weighted Imaging (DWI)Coronal Cine Balanced SSFP

The T2-weighted imaging (with and without fat suppression) is recognized as the pivotal MRE sequence for evaluating the small bowel, particularly for assessing active inflammation as intramural edema is best evaluated on fluid-sensitive T2-weighted sequences. Maximal slice thicknesses are suggested to be 5 mm for T2-weighted imaging. Motion-insensitive single-shot techniques are to be preferred as peristalsis-mediated artifacts are not attenuated by breath-hold or respiratory triggering techniques. Homogeneous fat suppression can be accomplished by several techniques, which include Dixon-based methods, chemically selective fat saturation, short tau inversion recovery, and adiabatic spectral inversion recovery [121,122]. FSE T2w sequences can be performed in either 2D or 3D, even if 2D should be preferred.

Balanced steady-state free precession gradient-echo (SSFPGR) sequences are recommended as they are relatively insensitive to motion artefacts and allow effective definition of the intestinal wall, as well as mesenteric structures, such as vessels and lymph nodes. It is also the sequence of choice in cine motility imaging. Maximal slice thicknesses are suggested to be 5 mm for SSFPGR imaging [122].

Gradient-echo (GRE) sequences and fast spin echo techniques can be used to accomplish T1-weighted MR imaging. T1w sequences should be performed in 3D. Three-dimensional GRE sequences enable fast acquisition time. Thus, imaging is obtained in most patients within the duration of 1 breath hold. In such a way, respiratory motion artefact is minimised. In addition, such sequences allow dynamic contrast enhancement through rapid repeated acquisitions.

Recently, it has been shown that radial 3D GRE sequences allow free-breathing T1-weighted imaging, which is particularly advantageous for patients who cannot hold their breath [123,124]. The maximum suggested slice thickness is 3 mm for both axial and coronal T1-weighted 3D sequences. Intravenous gadolinium should be administered with pump injection of 2 mL/s infusion rate, dose of 0.1–0.2 mmol/kg. Coronal FS 3D GRE should be performed pre- and post- contrast and the optimal timing of acquisition of sequences after injection can be in the enteric (45–50 s) or portal venous phase (60–70 s). An arterial phase can be performed, especially if bleeding is suspected as a complication. Axial FS 3D GRE delayed post contrast (90 sec/3–7 min) is particularly useful in identifying a pattern that can be ascribed to fibrosis. Contrast enhancement with gadolinium i.v. is particularly effective for the evaluation of penetrating disease so for example collections, abscesses and fistulas. In addition, post- contrast imaging is considered useful for showing mural inflammation and fibrosis [125,126]. Several researches have suggested that the use of post-gadolinium T1-weighted images increases diagnostic accuracy [127,128].

For all these reasons, the use of intravenous gadolinium is recommended by both European and US consensus guidelines when not contraindicated, as in the case of allergy or pregnancy. On the other hand, there is more and more evidence that the diagnostic accuracy can be maintained even without gadolinium administration, in particular if DWI is performed, and thus relying on T2-weighted sequences, DWI and cine motility sequences. Cine motility and DWI are suggested but not mandatory. European guidelines suggest that, when performed, DWI should be acquired in the axial plane (coronal plane is suggested, not mandatory), during free breathing and including b values ranging from 0 to 900 (b 0–50 and b 600–1000) and ADC map. The maximum slice thickness for DWI should be 5 mm. Axial acquisition is preferable to coronal acquisition because it is burdened by less artefacts. The known limitations of this sequence are given by the high sensitivity to magnetic susceptibility and motion artefacts. For this reason and to prevent misdiagnosis, DWI sequences should be interpreted alongside conventional sequences [129,130,131,132,133]. The specificity decreases if the bowel is not well distended. In fact, the loop not properly distended might show limited diffusion with high signal intensity on high b-value images and low signal intensity on the corresponding ADC maps: this can be a typical pitfall due to lack of loop distension. It is also important to remember that lymphoid hyperplasia is a frequent cause of false-positive results [134].

Cine sequences are achieved by performing rapid repeated slices through a single slice or region of interest using SSFPGR-based sequences [135,136,137]. Slice thickness is typically about 6–10 mm. The aim is to capture real-time intestinal peristalsis at high temporal resolution. The small intestine affected by IBD with inflammatory activity is characterised by altered motility, with reduced peristalsis movements [138]. Acquisitions can be targeted (e.g., on a loop with active signs of disease), but generally data are acquired from the entire small bowel volume by sequentially repeated coronal acquisitions at different anatomic regions. Cine MRI of motility is considered to be optional by both European and U.S. recommendations, although recent evidence suggests that cine imaging may improve diagnostic accuracy as well as aid assessment of disease activity and response to treatment [139,140,141,142,143]. For the evaluation of perianal fistulas and abscesses, MRE is superior to CTE due to its higher contrast resolution [117]. The pelvic MRE protocol dedicated to the evaluation of peri-anal fistula involves obtaining high-spatial-resolution small-field-of-view (15–20 cm) fat-suppressed and non-fat-suppressed T2-weighted, fat-suppressed contrast-enhanced T1-weighted, and diffusion-weighted images [120].

## 9. Imaging Findings

In this paper, we are going to review the imaging features associated with various stages of the same disease, from the uncomplicated acute phase, to the acute phase with complications arising from penetrating disease, to end with imaging features most associated with an inactive phase of the disease.

## 10. Imaging Findings Associated with Active CD Inflammation

As premised and previously mentioned, CD can affect the entire gastrointestinal tract, from mouth to anus, with alternation of healthy and pathological tracts. Imaging features that allow to distinguish pathologic involvement with signs of active inflammation involve both the small bowel loop and the mesentery consensually involved by pathology. With regard to loop pathology, typically, the loop involved by active pathology shows segmental mural hyperenhancement, wall thickening, intramural edema, and may be involved with stricture [110].

Segmental mural hyperenhancement is defined as an increase in mural signal intensity on MRE contrast material images assessed in an uncontracted small bowel segment and compared with the mural signal of a nearby normal small bowel segment [28]. Mural hyperenhancement can assume different aspects and hence may appear asymmetrical, stratified, or homogeneous. The appearance of asymmetrical mural hyperenhancement with predominant involvement of the mesenteric border of the loop is an imaging finding specific to small bowel CD [144,145]. We refer to stratified mural hyperenhancement when the submucosa is thickened by edema or inflammatory tissue (seen as high signal in T2w sequences) and the endoluminal side of the loop is characterised by increased contrast enhancement; this is called a bilaminar pattern. When thickening of the submucosa and endoluminal mucosa enhancement is associated with contrast-enhancing impregnation of the serosa as well, this is called a trilaminar pattern. The causes of stratification can be ascribed to a combination of factors, such as submucosal edema, inflammatory infiltration or fibrosis, the presence of granulation tissue, and intramural fat deposition [146]. For this very reason, the bilaminar and trilaminar aspects of the loop are also described in non-active disease with different signal characteristics in MRE than in the active pattern, which are discussed later among the signs of non-active disease [147]. More frequently, the trilaminar appearance is evidenced by MRE rather than CTE, and this is reasonably due to the higher intrinsic contrast resolution of MRE. Since the mucosa of inflamed intestinal segments is eroded and so missing on endoscopic evaluation and histopathological analysis, the term “mucosal hyperenhancement” is technically improper when the stratified enhancement pattern is visualized, and it would be appropriate to express it as “hyperenhancement of the endoluminal side” (Figure 1). Finally, the homogeneous symmetrical mural hyperenhancement, which is visualised in the images as a uniform transmural hyperenhancement of the entire intestinal wall, although found in patients with CD, is not distinctive of CD disease and may arise from other causes in which a varying degree of fibrosis, ischemia, fibro-adipose infiltration, or collagen deposition is present [148,149,150,151]. Concerning the technique, there are two phases in which it is possible to evaluate the wall enhancement: in the enteric phase (45–50 s after the beginning of the intravenous injection of contrast material) and in the portal venous phase (60–70 s after the beginning of the intravenous injection of contrast material) [122,152].

Wall thickness should be assessed in an adequately distended loop, thus with a caliper of at least 2 cm, and measured in the thickest portion of the most distended or most severely inflamed intestinal segment [153]. Wall thickening should be divided into mild (3–5 mm), moderate (>5–9 mm), or severe (≥10 mm) [154,155] (Figure 2). When bowel wall thickening is greater than 15 mm, and particularly if the thickening is asymmetrical or mass-like, neoplasia should be suspected, which may be concomitant with findings due to CD pathology per se [156,157,158].

Intramural edema, or mural edema, is detected as a hyperintense signal of the intestinal wall on fat-suppressed T2-weighted images or low b-value (i.e., a b value of 0–20 s/mm^2^) of DWI [135,159]. Intramural fat, which may be the result of previous intestinal inflammation, also shows high signal intensity on T2-weighted images, but differs from edema by losing signal intensity with fat-suppressed weighted sequences. Intramural edema is better evaluated on MRE because of its high contrast resolution than on CTE [41] (Figure 3).

A stricture is defined as a bowel segment with a luminal narrowing of at least 50% compared with that of an adjacent normal bowel loop with associated frank dilatation (≥3 cm) of the upstream bowel segment [160]. A stricture may be present with or without active inflammation, although most strictures have been shown to have an active inflammation component together with fibrosis. As a result of histopathologic analysis, the wall is frequently seen to consist of smooth muscle hypertrophy, although some degree of inflammatory cell infiltration or/and fibrosis may coexist [161,162,163,164,165]. It is important to describe the location and length of the stricture, whether it is at the level of the anastomosis in a patient who has undergone surgery or is a native stricture, whether the aforementioned signs of active inflammation are present, and whether upstream dilatation is present [166,167]. There is a strong association between stricture development and penetrating disease, so if a stricture with signs of active inflammation is present, it is important to look for the occurrence of an associated penetrating disease, such as a fistula, which normally arises in the middle or proximal aspect of a stricture. Just as if a penetrating disease is present, such as an inflammatory mass or fistula, it is important to consider that there may be a stenosis on a nearby intestinal segment, which usually has evidence of active inflammation [168,169,170]. Upstream loop dilatation may be mild (3–4 cm) or moderate to severe (>4 cm)(Figure 4). 

Ulceration is another finding associated with active inflammation. Ulceration consists in a rupture of the intra-luminal surface of the intestinal wall with extension of the endoluminal contents into the wall, a finding that is clearly visible in MRE with the use of oral contrast medium [171,172]. Unlike the sinus tract, which extends outside the wall to involve mesenteric fatty tissue, in the ulcer, the defect is confined to the bowel wall. Commonly known also as “penetrating ulcer”, this term should be avoided in the radiological report as it could create confusion with penetrating disease (e.g., a fistula or a sinus tract) or a pathological cardiovascular disease (Figure 5).

Sacculations, or pseudosacculations, refer to broad-based outpouching occuring along the antimesenteric border of an intestinal loop and are the manifestation of either acute or long-lasting inflammation of the intestinal wall and/or fibrosis. They are characteristically located along the antimesenteric border and correspond to asymmetric wall inflammation and mesentery hyperplasia along the mesenteric border [167,168,173,174]. The key sequences that allow to find the main signs of active inflammation are the T2w and DWI sequences: the two in concert allow to make diagnosis and to assess the severity of the pathological involvement [155,175].

As already mentioned, together with T2 and DWI, cine MRE can be helpful, especially in symptomatic patients who can only ingest a small volume of enteric contrast material. In fact, cine sequences, performed before the use of spasmolytic agents, can identify reduced bowel motility in a given bowel segment, thus helping to distinguish underdistended from inflamed bowel, increasing our diagnostic confidence in identifying bowel inflammation or stenosis [118,119,176]. In CD, the active inflammation of the intestinal wall is the cause of restricted diffusion of water molecules, so intestinal segments with restricted diffusion may present high signal intensity on high b-value of DWI (i.e., a b-value of at least 500 s/mm^2^) and low signal intensity on the corresponding apparent diffusion coefficient (ADC) maps [177] (Figure 6). Restricted diffusion is a non-specific sign. However, it is important to keep in mind that the segments do not always show high signal intensity and the signal can sometimes be intermediate, especially if the loops are not properly distended [178,179].

We have reviewed the imaging features affecting the bowel wall so far, but signs of acute inflammation are also consensually present in the mesentery. Characteristically, the mesentery may show signs of edema and inflammation, engorged vasa recta, fibrofatty proliferation and lymphadenopathy may be present.

Perienteric edema and/or inflammation, also known as “fat stranding” or “misty mesentery”, is visualised as increased T2 and DWI-weighted signal intensity on MRE images in the mesenteric fat adjacent to the diseased loops of bowel. Perienteric inflammation is consensual with wall inflammation because it often represents an extension of transmural inflammation of the intestinal wall itself [180].

Engorged vasa recta, also known as ‘comb sign’, are defined by the increased caliber of the vessels feeding and draining an inflamed intestinal loop [181,182]. The presence of recta vessels may be an expression of current intestinal inflammation, but also of previous inflammation (Figure 7).

Fibrofatty proliferation is a consequence of the inflammatory insult so the mesentery adjacent to diseased intestinal segments may undergo fibrous proliferation (sometimes also called “creeping fat” [182]. Fibrofatty proliferation, which is usually prominent in individuals who have experienced repeated episodes of acute inflammation, typically occurs on the mesenteric side but may also be circumferential. Hypertrophy of the fat tissue on the mesenteric side is often accompanied by an appearance of sacculation in the wall along the antimesenteric border. Mesenteric hypertrophy shows slightly decreased signal intensity on T1-weighted MRE images compared to normal fat, due to the influx of inflammatory cells and fluid [181,182].

Lymphadenopathies are usually present in the context of the mesenteric fat during acute inflammation. Mesenteric lymph nodes in CD are generally of reactive aetiology and enlarged lymph nodes (up to 1–1.5 cm in diameter on the short axis) are common (Figure 8).

Finally, during acute inflammation, thrombosis and/or occlusion of mesenteric venous vessels may occur, usually close to the inflamed intestinal segments [183]. It is important to specify in our radiological report whether the thrombosis has the characteristics of acute or chronic thrombosis, as this will determine whether or not the patient should be given anticoagulant therapy. Acute thrombosis is related with distension of the affected vein due to the presence of the endoluminal thrombus. In chronic occlusion, on the other hand, the central mesenteric veins may be narrowed or interrupted, so may be poorly visualised, while the collateral mesenteric vessels and/or small bowel varices are clearly evident and ectasic [183]. 

## 11. Imaging Findings Associated with Penetrating CD Inflammation and Complications

Penetrating CD includes sinus tract, simple fistula, inflammatory mass, abscess, and free perforation. These aspects are found in 25–33% of cases. In addition to defining whether intestinal inflammation and/or stenosis is present, it is always important to clearly specify in radiological reports whether penetrating CD is present, because it may require antibiotic treatment and/or drainage before administering immunosuppressive or biologic medications [184]. Fine irregularity of the serous side of the loop in a loop with characteristics that suggest active disease is a sign that should lead us to suspect a possible evolution towards penetrating disease.

A sinus tract is defined as a blind end tract that, contrary to what has been said for the ulcer, from the endoluminal side crosses the wall and extends beyond the serosa of the intestinal wall into the peri-visceral adipose tissue but does not reach the adjacent organs or the skin (Figure 9). If the pathology persists, the sinus tract becomes a true fistula.

A fistula is a pathway that connects the intestinal lumen to another epithelialized surface and are named after the structures they connect, such as an enteroenteric, entero-colic, recto-vaginal, enterocutaneous, or enterovesical fistula [185]. Fistulas usually arise from the middle or proximal aspect of a stenosis that usually has signs of active inflammation [185]. A simple fistula consists of a single extra enteric tract of communication, whereas a complex fistula results from the presence of more than one fistulous tract [186] (Figure 10). The complex fistula is the expression of advanced penetrating disease that heals and then reappears, thus creating complex structures. In fact, complex fistulas are associated with retraction phenomena as well as inflammation and fibrosis, and these processes result in the typical asterisk or cloverleaf or rosette shape, patterns highly suggestive of a complex fistula and that are given by the angle and the close connection that the intestinal loops involved assume [187]. Perianal fistulas are frequent in CD and originate from the rectal or anal canal, following active inflammation and deep ulceration of the mucosa. As with other internal fistulas, peri-anal fistulas may extend to the skin surface or any other epithelium-coated surface in the vicinity (e.g., urethra or vagina). The most validated classification systems for classifying fistulas are Park’s or St James’, which allow us to effectively indicate to the surgeon the type of fistula which is present [117,188]. In addition to the classification of the fistula, it is fundamental to indicate if the fistula is simple or complex and if there is an associated abscess because from those indications the management of the fistula varies considerably (Figure 11).

An inflammatory mass consists of dense mesenteric inflammation without a well-defined fluid component or discrete wall, occurring adjacent to an intestinal segment affected by mural inflammation [189]. An inflammatory mass exhibits variable signal intensity on MRE images, mixed with fat.

An abscess is a fluid collection bordered by well-defined thickened walls and characterised by contrast graphic enhancement on contrast-enhanced MRE, as opposed to an inflammatory mass. Internal gas may or may not be present within the fluid formation. Abscesses may be found within the mesentery, peritoneal cavity, retroperitoneum, body wall, or perirectal and/or perianal region (Figure 12). Abscesses usually in diffusion sequences show limited diffusion with high signal intensity on high b-value images (i.e., a b-value of at least 500 s/mm^2^) and low signal intensity on the corresponding ADC maps. As already mentioned, weighted DWI sequences represent one of the reference sequences for the evaluation of CD, and are fundamental in those patients with a contraindication to intravenous contrast material infusion [190]. Also in this case, the diffusion study has shown similar results compared with the post-contrast images regarding the complication’s detection. There are several studies evidencing that the T2w images associated with diffusion study can actually be used for the study of complications, even without the use of contrast [191,192].

Free perforation with the presence of intraperitoneal air is a rare complication of penetrating CD but should be precisely reported since it requires surgical evaluation [193,194] (Table 2). 

## 12. Imaging Findings Associated with Non-Active CD Inflammation

CD affecting intestinal loops can lead in the long term to permanent structural damage as a result of repeated inflammatory insults. The consequences are bowel wall alterations due to intramural fat deposition, various degrees of fibrosis, atrophy, and distortion of the intestinal loop [191]. The findings that are most suggestive of CD without active inflammation on imaging are the presence of fibrosis, fat deposition, pseudodiverticula/sacculations, pseudopolyps, and the absence of signs of active disease described above. In recent years, the traditional consideration that sees intestinal fibrosis in CD as an irreversible process has been progressively changing and nowadays fibrosis is seen as a chronic and progressive process but susceptible of reversibility, at least at an early stage of the disease course [191]. These recent findings have given impetus to the search for new antifibrotic agents, and the need to find new non-invasive and reliable imaging biomarkers that could quantify fibrosis and monitor its evolution, especially post-treatment. A pathologic intestinal segment without active signs of disease may show a thickened wall but decreased signal intensity on T2-weighted and DWI images, due to intramural adipose substitution [193,194] (Figure 13).

## 13. Conclusions

CD is a heterogeneous disease with multifactorial etiology. Imaging plays a key role in defining the disease, through the use of ultrasound, CTE, and MRE, with the latter representing the gold standard of study. In order to obtain an optimal examination, MRE must be performed through a rigorous protocol, with specific dedicated sequences, and preceded by adequate patient preparation. The radiologist’s effort must be directed to the evaluation of the disease, its extension, the possible associated complications, and the degree of activity, with the aim of guiding the most appropriate medical and/or surgical treatment for each type of patient.

## Figures and Tables

**Figure 1 diagnostics-12-01236-f001:**
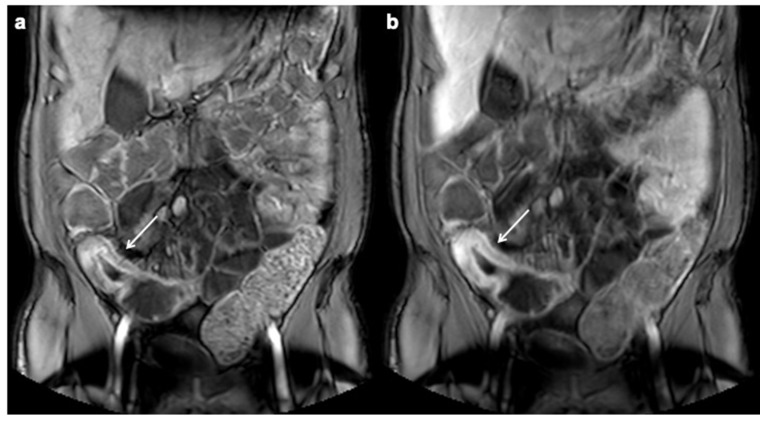
Segmental mural hyperenhancement. Coronal contrast enhanced T1-weighted MRE images in the enteric (**a**) and portal venous (**b**) phases show small bowel wall thickening with trilaminar mural hyperenhancement (white arrows), findings consistent with active inflammatory CD.

**Figure 2 diagnostics-12-01236-f002:**
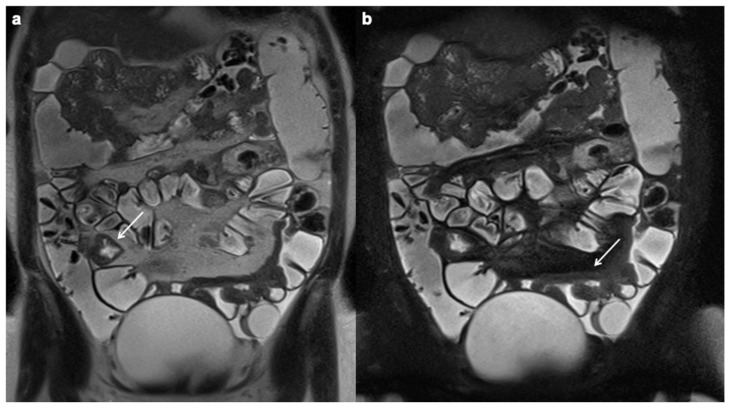
Moderate wall thickening (9 mm). Coronal T2 FSE without FS (**a**) and coronal T2 FSE with FS (**b**) MRE images demonstrate moderate small bowel wall thickening in right flank and pelvic cavity (white arrows), free fluid is associated in the left iliac fossa.

**Figure 3 diagnostics-12-01236-f003:**
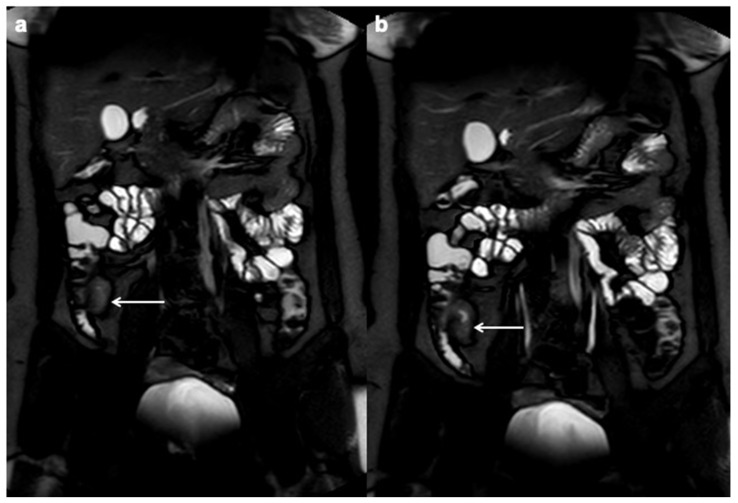
Intramural edema. Coronal T2 FSE with FS MRE images (**a**,**b**) in a patient with CD detect wall thickening in the distal ileum, with increased mural signal intensity (white arrows), finding consistent with edema.

**Figure 4 diagnostics-12-01236-f004:**
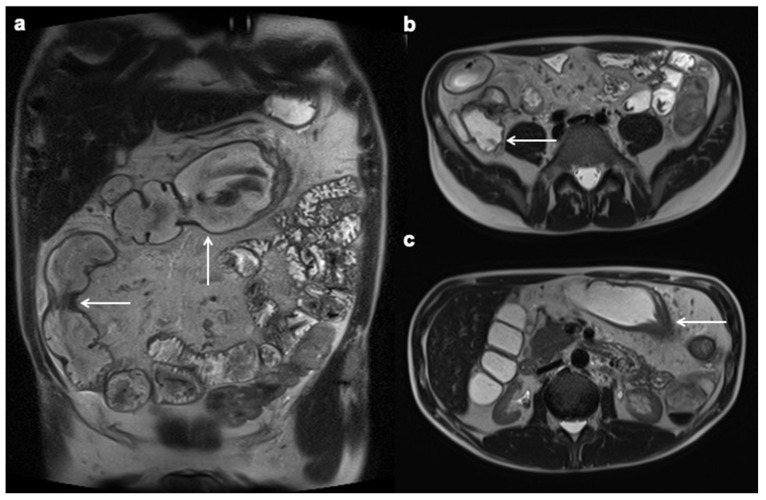
Stricture with upstream dilation (white arrows). Coronal (**a**) and axial (**b**,**c**) T2 FSE without FS MRE demonstrate ileal and transverse colon stricture with respectively mild (3–4 cm) and severe (>4 cm) upstream dilation.

**Figure 5 diagnostics-12-01236-f005:**
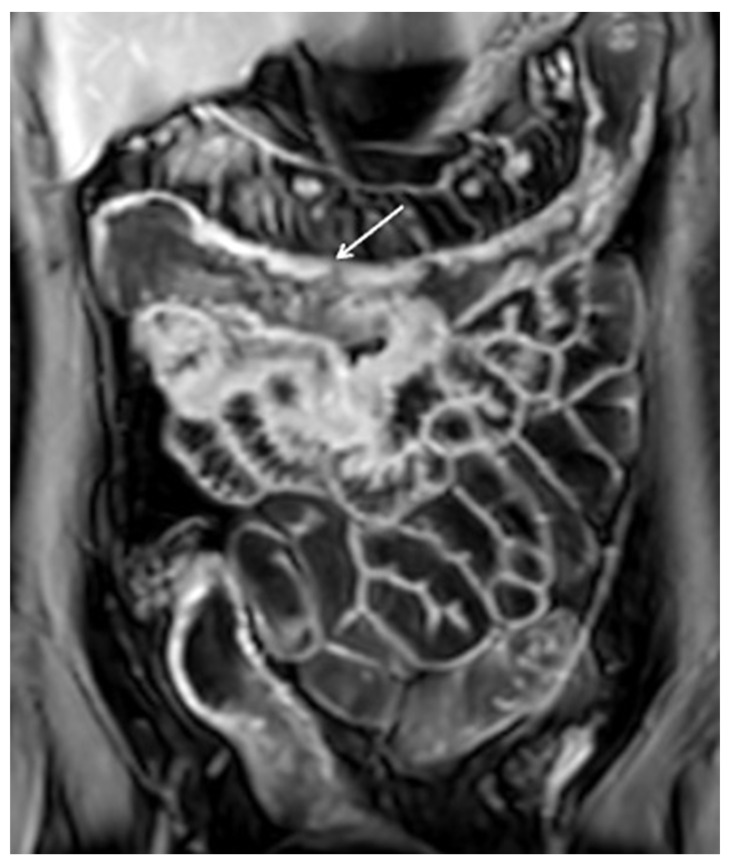
Ulcerations. Coronal contrast-enhanced fat-suppressed T1-weighted MRE portal venous phase image shows transverse colon with mural hyperenhancement and multiple intramural penetrating ulcers (white arrow), findings consistent with active inflammatory CD.

**Figure 6 diagnostics-12-01236-f006:**
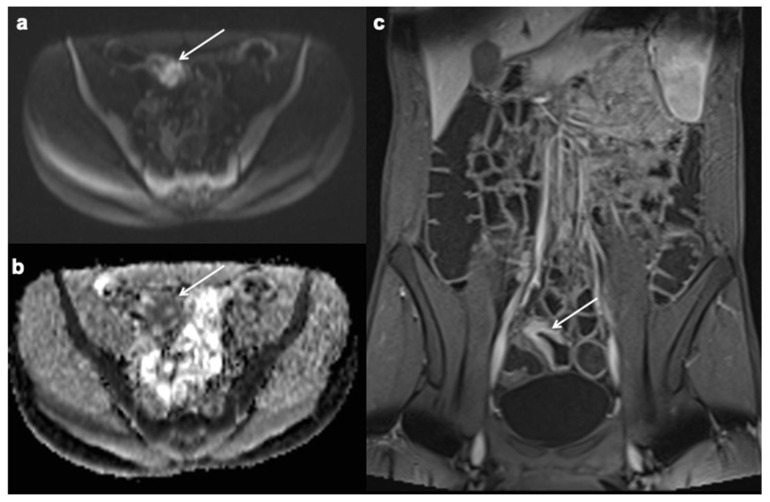
Restricted diffusion. Axial diffusion-weighted (b = 800 s/mm^2^) (**a**), ADC map (**b**) MRE images in patient with CD detect marked bowel wall diffusion restriction in the terminal ileum. This intestinal segment has hyperintense signal (arrow in (**a**)) on the diffusion-weighted image, hypointense signal (arrow in (**b**)) on the ADC map and mural hyperenhancement in contrast-enhanced fat suppressed T1- weighted portal venous phase (**c**) (white arrows).

**Figure 7 diagnostics-12-01236-f007:**
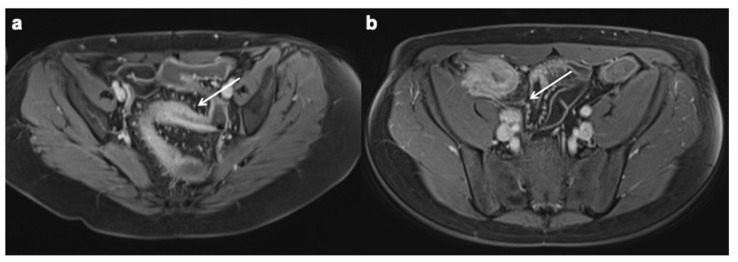
Engorged vasa recta. Axial contrast-enhanced fat-suppressed T1 weighted portal venous phase (**a**,**b**) MRE images show engorged vasa recta (white arrows) in the small bowel mesentery owing to active inflammatory CD.

**Figure 8 diagnostics-12-01236-f008:**
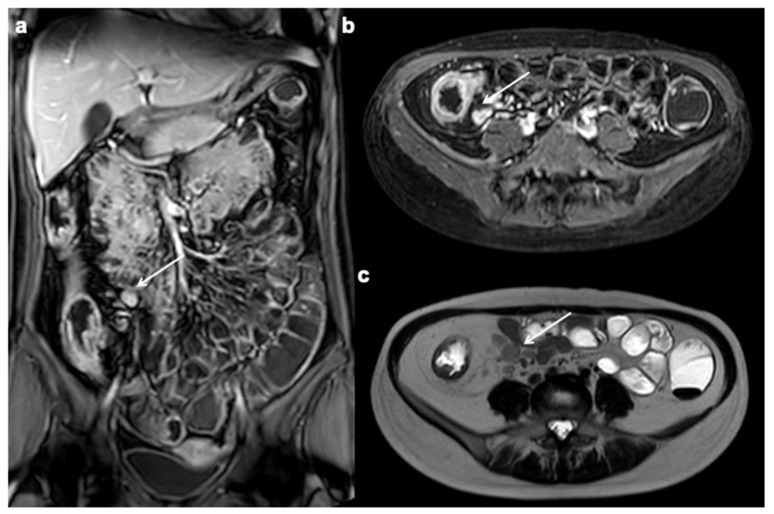
Lymphadenopathy. Coronal, axial contrast-enhanced fat-suppressed T1 weighted portal venous phase (**a**–**c**) and axial T2 FSE without FS show a lot of mesenteric enlarged lymph nodes (up to 1–1.5 cm in diameter on the short axis) (white arrows).

**Figure 9 diagnostics-12-01236-f009:**
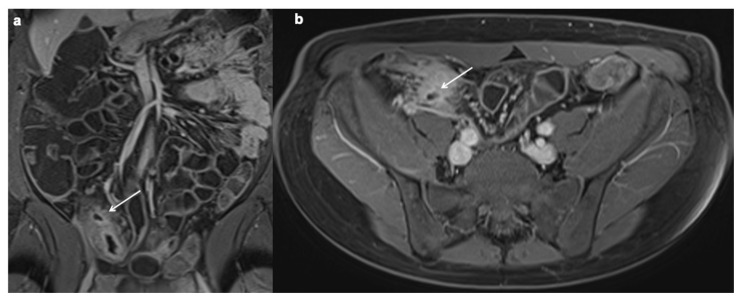
Sinus tract. Coronal and axial contrast-enhanced fat-suppressed T1 weighted portal venous phase (**a**,**b**) MRE images show a sinus tract that from the endoluminal side crosses the wall and extends beyond the serosa of the intestinal wall into the peri-visceral adipose tissue (white arrows).

**Figure 10 diagnostics-12-01236-f010:**
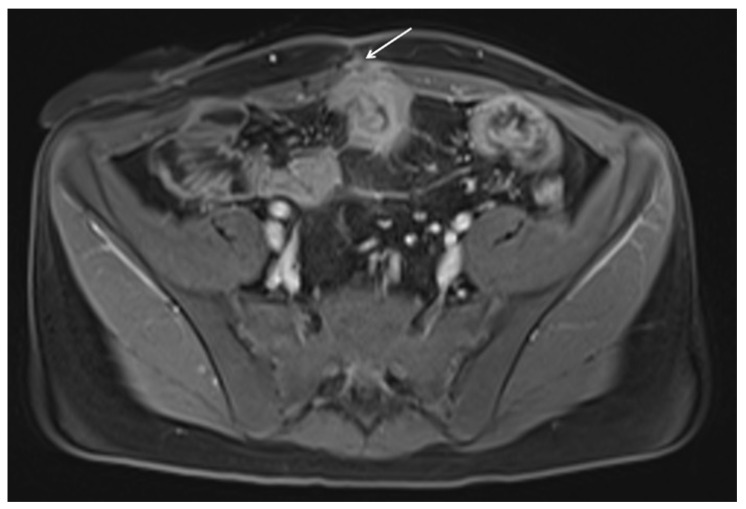
Axial contrast-enhanced fat-suppressed T1 weighted portal venous phase MRE images shows a severe small bowel thickening with an associated enteorcutaneous fistula (white arrows).

**Figure 11 diagnostics-12-01236-f011:**
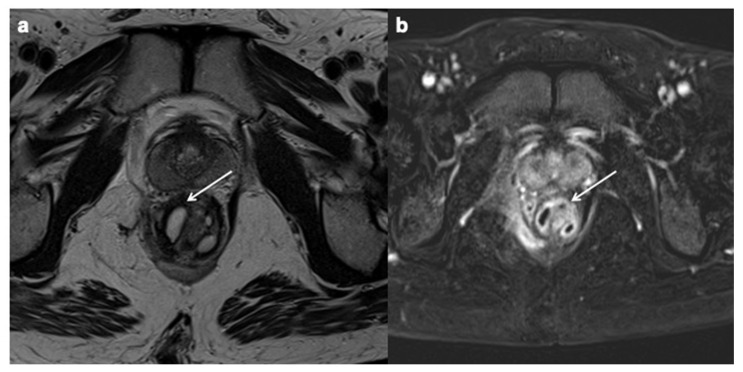
Perianal CD (**a**,**b**). Axial T2 FSE without FS and axial contrast-enhanced fat-suppressed T1 weighted enteric phase MRE images (white arrows) show a intersphinteric perianal fistula with horseshoe appearance.

**Figure 12 diagnostics-12-01236-f012:**
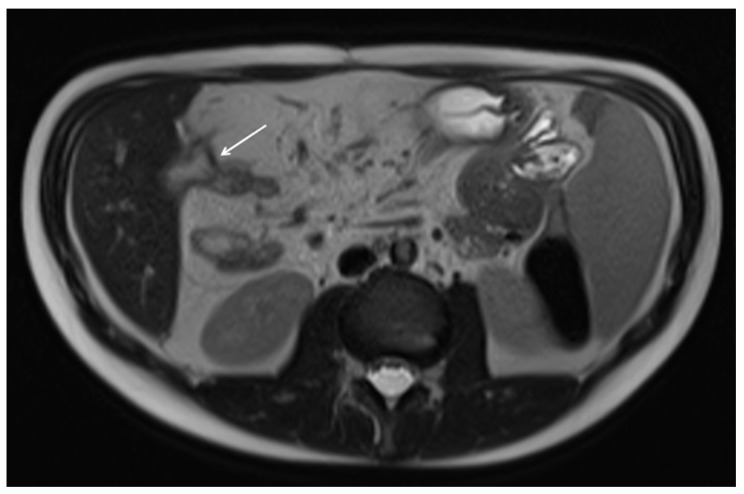
Axial T2 FSE without FS MRE image demonstrates a peri-hepatic abscess in a patients with active CD (white arrows).

**Figure 13 diagnostics-12-01236-f013:**
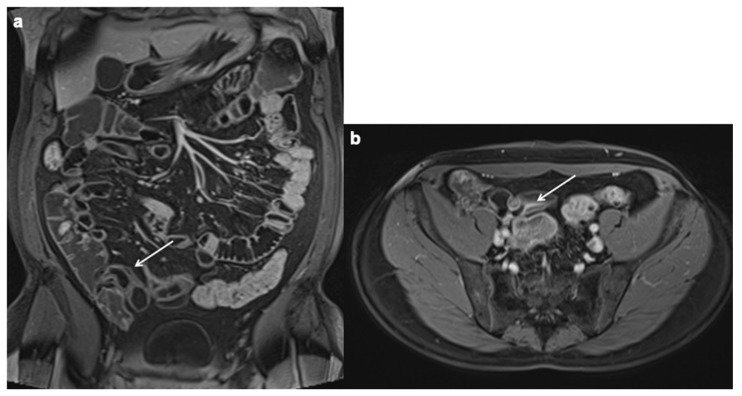
Coronal and axial contrast-enhanced fat-suppressed T1 weighted enteric phase (**a**,**b**) detect small bowel wall thickening in the terminal ileum with intramural fat deposition (white arrows), representing no signs of active inflammation.

**Table 1 diagnostics-12-01236-t001:** Small bowel magnetic resonance imaging protocol for 1.5 T.

Sequence	Orientation	TR (ms)	TE (ms)	2D/3D	ST (mm)	FS
TRUFI T2-W BH	Coronal	3.45	1.46	2D	4	without
TRUFI T2-W BH	Axial	3.73	1.87	2D	4	without
TRUFI T2-W BH 20 measures (motility study)	Coronal	3.57	1.79	2D	10	without
HASTE T2-W BH	Coronal	600	87	2D	5	without
HASTE T2-W BH FS	Coronal	500	87	2D	5	with (SPAIR)
HASTE T2-W BH	Axial	500	87	2D	5	without
HASTE T2-W BH FS	Axial	500	87	2D	5	with (SPAIR)
DWI ep2d diff 3 av (b values 0.600; 3 averages)	Axial	2500	85	2D	5	with
VIBE T1-W FS precontrast	Axial	3.24	1.1	3D	3	with (SPAIR)
VIBE T1-W FS postcontrast	Coronal	3.24	1.1	3D	3	with (SPAIR)
VIBE T1-W FS postcontrast	Axial	4.89	2.39	3D	3	with (SPAIR)

TR = repetition time; TE = echo time; ST = slice thickness; FS = fat suppression; BH = breath hold; TRUFI = true fast imaging with steady-state free precession; HASTE = half Fourier single-shot turbo spin-Echo; SPAIR = Spectral adiabatic inversion recovery; DWI ep2d diff 3 av = diffusion weighted imaging, echo planar imaging 2d multislice diffusion imaging, 3 averages; VIBE = volumetric interpolated breath-hold examination.

**Table 2 diagnostics-12-01236-t002:** Imaging findings associated with active and complicated CD.

Imaging Findings Associated with Active CD Inflammation
Segmental mural hyperenhancement - *Asymmetric* - *Stratified* - *Homogeneous*
Wall thickening - *Mild (3–5 mm)* - *Moderate (>5–9 mm)* - *Severe (**≥10 mm)*
Intramural edema
Stricture
Ulcerations
Sacculations
Perienteric edema and/or inflammation
Engorged vasa recta
Fibrofatty proliferation
Mesenteric venous thrombosis and/or occlusion
Lymphadenopathy
Restricted diffusion
Diminished motility
**Imaging findings associated with penetrating CD and complications**
Sinus tract
Fistula - *Simple* - *Complex*
Inflammatory mass
Abscess
Free perforation

## Data Availability

All data are reported in the manuscript.

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
