# Peer review of "The Role of Magnetic Resonance Enterography in Crohn’s Disease: A Review of Recent Literature"

_diagnostics, 2022, doi:10.3390/diagnostics12051236_

Round 1

Reviewer 1 Report

There are many reviews on MR ENTEROGRAPHY IN Crohn's disease in literature

After going through your review, it is more like encyclopedia of MR Enterography in Crohn's disease.

Review is too long , needs to be curtailed

Review should be critical one

Author Response

Thank you for the revision, we have significantly shortened the text, I hope it can now be more appropriate. 

Reviewer 2 Report

The paper is a review of the role of imaging scans, and especially MR enterography in the diagnosis of Crohn's disease (CD).

Although the examination technique and imaging findings are described in detail, many aspects of the role of MRI in CD are not described: e.g., the role in the therapeutic response,  the role in the assessment of postoperative recurrence, and scorification.

The generalities regarding CD and the rest of the imaging methods are too long and unrelated to the subject of the paper.

I would also propose a better systematization, a table that highlights the main MR features for diagnosis.

It would also be useful to list the main studies on the role of MRI in CD.

Given that there are already published numerous reviews and consensus on this topic, the authors should emphasize what are the strong points of their paper that would justify the publication.

Author Response

Thank you for the review, we have added the requested missing sections, a summary table of rm features of Crohn's and shortened the text where redundant, I hope the article can be more appropriate now.

Reviewer 3 Report

Comments to the Author

Title: The role of Magnetic Resonance Enterography in Crohn's disease: a review of recent literature

Authors: Marysol Biondi, Eleonora Bicci, Ginevra Danti, Federica Flammia, Giuditta Chiti, Pierpaolo Palumbo, Federico Bruno, Alessandra Borgheresi, Roberta Grassi, Francesca Grassi, Roberta Fusco, Vincenza Granata, Andrea Giovagnoni, Antonio Barile and Vittorio Miele

à The paper provides a review of recent literature regarding the role of Magnetic Resonance Enterography (MRE) in Crohn's disease (CD). The manuscript aims to prove the role of imaging in the diagnosis and follow-up of patients with CD using MRE and includes a description of the main imaging features during the various phases of disease activity and its complications.

à The study reported in the manuscript is interesting and well presented. I would endorse the publication of the present manuscript after carefully addressing the comments hereafter.

General: The manuscript is well written however, some sections need some more work.

In the following: MA=Major comment, MI = Minor comment, OP = Optional Comment

 (MI) P2, Abstract, L75-76: I think the word ‘Abstract’ should appear in bold but no ‘Inflammatory’.

(OP) P2, Abstract, L79-81: These lines are redundant with the previous ones. I suggest deleting them and starting the new one as: The classic CD signs are…

(MI) P3, Introduction, L95: Include the UC abbreviation in the abstract.

(MI) P3, Introduction, L107: To avoid redundancies try to combine this sentence with sentence L103.

(MI) P3, Introduction, L114 -115: Do you mean UC and CD when saying in both pathologies? Please clarify.

(MI) P3, Introduction, L119: Introduce what the acronym ESGAR stands for.

(MI) P3, Introduction, L134: Healthy segments of what? Please clarify the organ under study.

(MI) P4, Introduction, L160: Please, check with the author guidelines if this is the correct way for referencing. I believe you are missing the dot and the period in et al.

(OP) P4, Introduction, L176: Consider including a reference here.

(OP) P5, CD, L197-202: Consider including more reference in this paragraph.

(MI) P5, CD, L227: What is the percentage of patients that need surgery? And is there any mortality rate associated with CD?

(OP) P5, Role of Imaging, L235: Instead of these patients I would write CD patients.

(OP) P5, Role of Imaging, L239-241: These sentences are wordy. Please re-write in a better way.

(OP) P5, Role of Imaging, L239-241: These sentences are wordy. Please re-write in a better way.

(MI) P5, Role of Imaging, L241: Other techniques such as US should also be mentioned in this sentence.

(MI) P6, Role of Imaging, L280: Add reference.

(MI) P6, Role of Imaging, L280: Add reference.

(MI) P6, Role of Imaging, L285: What is the typical dose range received by a patient in a CT scan? How does this value compare with exposure limits?

(MA) P7, Role of Imaging, L297: The authors should include some basic principles about CT imaging and the type of scanners used for CD.

(MI) P7, MRE, L300-304: This has been mentioned several times in the manuscript. Please, avoid redundancies to enhance reading fluidity.

(MA) P7, MRE, L299-328: I found this section redundant (in particular the first paragraph). I would like the authors to rework this section trying to avoid repeating the same information. They should include some basic principles about MRI imaging.

(MI) P7, General patient preparation, L340: Add references.

(MI) P7, General patient preparation, L340-341: This information is the same one than the previous sentence.

(MI) P8, General patient preparation, L362: Add references.

(MI) P8, General patient preparation, L368-369: What is the typical dose range received by a patient during the Fluoroscopy guidance?

(MA) P8, General patient preparation, L383: The authors should include a table summarizing the techniques used in CD diagnosis and monitoring. The table should include the advantages and disadvantages of each technique.

(MI) P8, Technical Considerations and Sequence Selection, L385: Mention that this is for MRE. Non-experts may be confused by 1.5 and 3 T and what does it mean.

(MA) P8, Technical Considerations and Sequence Selection, L394: These sequences are for CD diagnose or abdominal evaluation in general? Please clarify.

(MI) P8, Technical Considerations and Sequence Selection, L394: These sequences are for CD diagnose or abdominal evaluation in general? Please clarify.

(MI) P9, Technical Considerations and Sequence Selection, L414: Add reference for SSFPGR.

(OP) P10, Technical Considerations and Sequence Selection, L469: The information of this section will be easier to follow if presented in a table or a different format. I encourage the author to include a table or diagram to better shown the information provided.

(MI) P11, Imaging findings, L500-501: There is a typo, the sentence appears split in two different lines.

(MI) P11, Imaging findings, L500-501: There is a typo, the sentence appears split in two different lines.

(MI) P11, Imaging findings, Figure 1: The authors need to mention the image source.

(MI) P12, Imaging findings, Figure 2: The authors need to mention the image source.

(MI) P12, Imaging findings, L542: There is a typo, the 2 in mm2 should be superindex.

(MI) P13, Imaging findings, Figure 3: The authors need to mention the image source.

(MI) P12, Imaging findings, 370: How likely is for the patient to develop the bacteria infection?

(MI) P12, Imaging findings, 572-575: It would be useful if the author provide statistical values for each mentioned case.

(MI) P14, Imaging findings, Figure 4: The authors need to mention the image source.

(MA) P14, Imaging findings, 590: The authors should describe what is the image showing in the text. This way the reader knows beforehand the information that is being provided by the Figure and its context with the section.  The same applies for all previous Figures.

(MI) P14, Imaging findings, 596: Remove the word also.

(MI) P15, Imaging findings, Figure 5: The authors need to mention the image source.

(MI) P16, Imaging findings, L616: There is a typo, the 2 in mm2 should be superindex.

(MI) P16, Imaging findings, Figure 6: The authors need to mention the image source. Again, I encourage the authors to better introduce Figures within the context of the text.

(MI) P17, Imaging findings, Figure 7: The authors need to mention the image source.

(MI) P18, Imaging findings, Figure 8: The authors need to mention the image source.

(MA) P18, Imaging findings, L669: I believe the authors should summarize the information contained in this section in a Table including the main discussed characteristics such as complications or most suitable MR sequences.

(MI) P18, Imaging findings, L671: How likely is to develop penetrating CD? Do you have any quantitative information?

(MI) P19, Imaging findings, Figure 9: The authors need to mention the image source.

(MI) P19, Imaging findings, L691: There is a typo, a space is missing after the dot in ‘192].A simple’

(MI) P20, Imaging findings, Figure 10 and 11: The authors need to mention the images source and better introduce them in the text.

(MI) P21, Imaging findings, L714: clarify why the term phlegmon is misleading.

(MI) P21, Imaging findings, L725: There is a typo, the 2 in mm2 should be superindex.

(MI) P21, Imaging findings, L731: Introduce and explain what is mdc.

(OP) P21, Imaging findings, L732-733: Only for a senior radiologist? What about a novel one? If so, why are these differences in diagnostic happening?

(MI) P21, Imaging findings, Figure 12: As for all previous Figures, the authors need to mention the image source.

(MI) P21, Imaging findings, L739: As mentioned for the previous sub-section, the authors should consider including all the information provided in a Table.

(MI) P22, Imaging findings, L745: Add reference.

(MI) P22, Imaging findings, L773: If the patients were adults, will the outcomes be different?

(MI) P21, Imaging findings, L785: There is a typo, the 2 in mm2 should be superindex as well as the -1 in s-1.

(MI) P22, Imaging findings, L800: In the previous page, the noun seconds has been abbreviated as s and here as sec, please be consistent with the units.

(MA) P23, Imaging findings, L836: The following paragraphs should be included in a new section called: Future trends or something similar.

The authors should provide a better outlook, include future research and research that is already on going. Should also specify what is missing and what needs to be addressed…

(MI) P24, Imaging findings, Figure 13: The authors need to mention the image source.

(MA) P24, Imaging findings, L858: As for the ‘Future trends’ section, the authors should include a conclusion section.

The conclusion provided is weak, more work needs to be done for this section. I miss a concluding remarks section in which the authors summarize in a concise manner all the information provided. Also, since this is a review article, the authors should provide a time line of the evolution of MRE in CD.

(MA) P24-33, References: The reference list is too long, even for a review. Please try to avoid referencing complementary works.

Author Response

Thank you for the timely review of the manuscript and the many suggestions for improvement of the same. We have shortened the manuscript, reduced the extent the number of references, added summary tables for both the rm characteristics of CD and the study technique. We have made all requested changes in the new text. All images shown are original and are images of our cases from our department.
Thank you, I hope the changes are appropriate, we look forward to feedback.

Round 2

Reviewer 2 Report

The authors have improved their work

Reviewer 3 Report

Dear authors thank you for considering and including all my comments and suggestions,

I endorse the publication of the present manuscript.

Thank you,